# Integrative Analysis of Transcriptome, Proteome, and Phosphoproteome Reveals Potential Roles of Photosynthesis Antenna Proteins in Response to Brassinosteroids Signaling in Maize

**DOI:** 10.3390/plants12061290

**Published:** 2023-03-13

**Authors:** Hui Li, Xuewu He, Yuanfen Gao, Wenjuan Liu, Jun Song, Junjie Zhang

**Affiliations:** 1Institute of Quality Standard and Testing Technology Research, Sichuan Academy of Agricultural Sciences, Chengdu 611130, China; 2College of Life Science, Sichuan Agricultural University, Ya’an 625014, China

**Keywords:** transcriptome, proteome, phosphoproteome, photosynthesis antenna proteins, brassinosteroids, signaling, differentially expressed genes, differentially expressed proteins

## Abstract

Brassinosteroids are a recently discovered group of substances that promote plant growth and productivity. Photosynthesis, which is vital for plant growth and high productivity, is strongly influenced by brassinosteroid signaling. However, the molecular mechanism underlying the photosynthetic response to brassinosteroid signaling in maize remains obscure. Here, we performed integrated transcriptome, proteome, and phosphoproteomic analyses to identify the key photosynthesis pathway that responds to brassinosteroid signaling. Transcriptome analysis suggested that photosynthesis antenna proteins and carotenoid biosynthesis, plant hormone signal transduction, and MAPK signaling in CK VS EBR and CK VS Brz were significantly enriched in the list of differentially expressed genes upon brassinosteroids treatment. Consistently, proteome and phosphoproteomic analyses indicated that photosynthesis antenna and photosynthesis proteins were significantly enriched in the list of differentially expressed proteins. Thus, transcriptome, proteome, and phosphoproteome analyses showed that major genes and proteins related to photosynthesis antenna proteins were upregulated by brassinosteroids treatment in a dose-dependent manner. Meanwhile, 42 and 186 transcription factor (TF) responses to brassinosteroid signals in maize leaves were identified in the CK VS EBR and CK VS Brz groups, respectively. Our study provides valuable information for a better understanding of the molecular mechanism underlying the photosynthetic response to brassinosteroid signaling in maize.

## 1. Introduction

Maize (*Zea mays*) is an important C4 crop that greatly contributes to ensuring global food security [1,2]. Therefore, photosynthesis is recognized as playing a key role in improving crop yield; furthermore, ear photosynthesis may be particularly important for the final yield [3,4]. 

A range of stress factors are well known to negatively influence maize yield because of their effects on photosynthesis [5,6]. In this respect, hybrid maize varieties show a higher yield compared to the parent lines because of the more stable photosystem II and higher density of thylakoids in mesophyll chloroplasts [7,8,9]. Several studies have shown that photosystems are crucial in plants that respond to shade and high population densities [10,11,12]. Further, epibrassinolide (EBR), a bioactive brassinosteroid with favorable safety and commercial availability, and brassinazole (Brz), a BR biosynthesis inhibitor, play a crucial role in plant growth and development [13,14]. A previous study showed that exogenously applied BRs affect various aspects of photosynthesis and effectively improve plant yield [15]. Brassinosteroids (BRs) are a group of plant hormones that regulate various aspects of plant growth and development, including cell elongation, division, differentiation, and responses to biotic and abiotic stresses; BRs also interact with other hormones [16]. BES1/BZR1 (BRI1 EMS SUPPRESSOR 1/BRASSINAZOLE RESISTANT 1) is a central transcription factor (TF) in the BR signaling pathway that has been extensively studied in plants. In maize, there are eleven BES1/BZR1 members, most of which are basic helix–loop–helix (bHLH) TFs, while ZmBES1/BZR1-4 and ZmBES1/BZR1-5 have an additional β-amylase (BAM) catalytic domain at their C-terminal [17,18]. ZmBES1/BZR1-5 was reported to respond to osmotic stress in plants as well as form a dimer via its BAM domain, interacting with casein kinase II subunit β4 (ZmCKIIβ4) and ferredoxin 2 (ZmFdx2) to positively regulate kernel size in maize [19,20]. Recently, the crystal structure of BAM (ZmBES1/BZR1-5) was determined, providing insights into its noncatalytic adaptation and suggesting that BZR1-type BAMs may act as putative regulatory domains and/or metabolic sensors and dual-function TF in plants [21]. These diverse core TFs may act as a central hub in the BR signaling pathways and crosstalk with other hormones. However, our understanding of the mechanisms underlying photosystem responses to BR signaling in maize leaves remains fragmentary.

Photosynthesis converts solar energy into chemical energy and splits water molecules into oxygen, thereby supporting aerobic life [22]. The most important plant organs responsible for photosynthesis are the leaves, which have over 100 chloroplasts in each mesophyll cell [23]. Chloroplasts harbor the photosynthetic machinery, which is responsible for light harvesting and energy conversion. There are two photosystems within chloroplasts, namely, photosystem I (PSI), which is composed of the light-harvesting complex I (LHCI), and photosystem II (PSII), which is composed of light harvesting complex II. Both PSI and PSII have a peripheral antenna system [24]. There are five Lhca (a stands for LHCI) genes which encode LHCI proteins, and six Lhcb (b stands for LHCII) genes which encode LHCII proteins, present in PSI and PSII, respectively [25]. The antenna system is associated with light saturation problems and is an important regulator in PSI and PSII [26]. However, the regulatory mechanism of the photosynthesis antenna system during BR signaling remains unclear.

Several studies have shown that chlorophyll and light-harvesting chlorophyll b-binding protein (lhcb) play a central role in plant photosystems [27,28,29]. As is well known, lhcb and chlorophyll harvest photons and transport electrons to the biochemical energy center [30]. Numerous studies have shown that the total leaf chlorophyll content increases in various plant species, such as mung bean [31], Indian mustard [13,32], chickpea [33], geranium [34], and maize [35], under abiotic stress. Another study showed that in faba bean, abiotic stress had a positive effect on chlorophyll fluorescence parameters [36]. 

Brassinosteroids regulate photosynthesis both transcriptionally and by post-transcriptional modifications [37,38]. Here, we aimed to fully understand the mechanism whereby exogenously applied BR affect photosynthesis at various levels. Transcriptomes, proteomes, and phosphoproteomes are useful tools for the identification of RNA transcripts, proteins, and phosphopeptides, respectively [39]. Several studies have already used transcriptomics, proteomics, and phosphoproteomics to understand signaling mechanisms and have identified a large number of hormone-responsive genes, but these techniques have scarcely been used in photosynthesis studies [40,41,42]. Transcriptomes generate millions of reads and provide useful information on digital gene expression, alternative splicing, and nucleotide variation. Modern proteomics techniques allow identifying proteins that are differentially synthesized in response to hormones. At the same time, protein phosphorylation is also a key factor in hormone regulation networks. Therefore, we used phosphoproteomic analysis to reveal the diversity of activated signaling pathways during photosynthesis. The mRNA expression levels usually do not directly match the corresponding proteins and phosphopeptides levels. However, combined analysis of the levels of mRNA expression, proteins, and phosphopeptides can generate complementary data to determine their integrated biological function [43,44].

Here, we used high-throughput mRNA sequencing (RNA-Seq) and isobaric tags for relative and absolute quantification (iTRAQ) to identify and quantify the transcriptomes, proteomes, and phosphoproteomes of maize leaves with different hormonal status. Leaves of the maize inbred line Mo17 were exposed to EBR and Brz treatments. We compared mRNA, protein, and phosphopeptide expression profiles of control (CK) VS EBR-, CK VS Brz-, and EBR VS Brz-treated leaves. Differentially expressed genes (DEGs), proteins (DEPs), and phosphopeptide functions were analyzed using gene ontology (GO) and the Kyoto Encyclopedia of Genes and Genomes (KEGG) databases. We analyzed the different pathways and DEGs upon hormone treatment to determine the association between BRs and photosynthesis in maize. 

## 2. Results

### 2.1. Genome-Wide Expression Changes in mRNA, Protein Expression, and Phosphoproteins in Response to Brassinosteroid Treatment

To understand the regulatory mechanisms of BRs and explore new regulators, brassinosteroid- and brassinazole-responsive transcriptomes, proteomes, and phosphoproteomic changes in maize leaves were studied using RNA-seq and iTRAQ. Three samples (CK, EBR, and Brz) with three biological replicates each were used for transcriptome analysis. Nine cDNA libraries were prepared from the samples and subjected to paired-end sequencing. Gene expression was calculated as fragments per kilobase per million reads (FPKM). For RNA-seq, each library ranged from 38.15 to 50.27 million reads. Approximately 87% (86.34–88.01%) of clean reads for each library from the RNA-seq analysis were successfully mapped to the maize genome (Table 1).

The leaf proteomes were quantitatively catalogued using iTRAQ. The samples were divided into ten fractions; then, the search algorithm Mascot was used to identify proteins, and the results were filtered with a 1% false discovery rate, FDR [45]. A grand total of 74,882 MS/MS spectra were generated, and 16,968 spectra were matched to the secondary MS/MS spectra database. A total of 13,027 peptides were identified, 10,988 of which were unique peptides, while 3466 were mapped in the database (Figure 1A).

The MS data for phosphorylation enrichment were identified using Mascot software. In all, 3029 peptides and 2202 phosphatide peptides were identified in the MS data. In addition, 1681 proteins and 1066 phosphoproteins were mapped after filtering based on an FDR cut-off of 1% (Figure 1B). 

### 2.2. Transcriptome Differences in Mo17 in Response to BR and Brz Treatments

Gene expression was examined using the R package DEseq2 to quantify and analyze all control and treatment combinations. After 24 h of hormone exposure, 2908 genes were differentially expressed in Brz-treated and CK leaves, of which 1199 and 1709 were up- and downregulated, respectively. There were 661 genes that were differentially regulated in EBR-treated and CK leaves, of which 430 were up-, and 231 were downregulated. In turn, a total of 1871 DEGs were identified between the BR and Brz treatment groups, including 628 up- and 1243 downregulated genes (Table 2).

These DEGs were compared using KEGG for enrichment analysis to determine the biological pathways involved in BR signaling. The enriched terms for the DEGs between CK and EBR, CK and Brz, and EBR and Brz were mainly associated with the MAPK (mitogen-activated protein kinase) signaling pathway and plant hormone-signal transduction, both of which were previously shown to play important roles in BR signal [46,47]. Further, there were 23 DEGs involved in the MAPK signaling pathway and 34 in plant hormone-signal transduction in CK VS Brz, 18 DEGs in the MAPK signaling pathway and 26 in plant hormone-signal transduction in EBR VS Brz, and only 8 DEGs in plant hormone-signal transduction in CK VS EBR (Figure 2A).

Subsequent GO enrichment analysis, conducted to identify the plant systems influenced by the EBR and Brz treatments, showed that the DEGs were categorized into three groups: cellular components (CC), molecular function (MF), and biological processes (BP). The enriched terms for the DEGs between CK VS EBR and between CK VS Brz indicated the plasma membrane, cell periphery, DNA binding transcription factor activity, and hormone signaling (Figure 2B).

### 2.3. Proteome Changes in Mo17 in Response to BR and Brz Treatments

DEPs were analyzed using a total of 3466 proteins identified in both lines. A threshold of FC (fold change) ≥ 1.2 for upregulated proteins and FC ≤ 0.833 for downregulated proteins, together with a *p*-value ≤ 0.05 were chosen. Based on these criteria, 251 proteins were identified as differentially expressed between CK and Brz-treated leaves, of which 99 were up- and 152 were downregulated. Additionally, 297 DEPs were detected between CK and EBR, of which 128 were up- and 169 were downregulated. Finally, a total of 363 DEPs were identified between EBR and Brz, including 202 up- and 161 downregulated proteins (Figure 3A). 

The Kyoto Encyclopedia of Genes and Genomes is a useful database for understanding the high-level functions and utilities of biological systems. Usually, genes within the same pathway cooperate with each other to perform their biological functions [48]. The KEGG pathways enriched in CK VS Brz, CK VS EBR, and EBR VS Brz were mainly focused on ribosomes and photosynthesis. Furthermore, specific enriched pathway for CK VS Brz were benzoxazinoid biosynthesis, metabolic pathways, phagosomes, and starch and sucrose metabolism. Meanwhile, for CK VS EBR, the enrichment terms included phagosome and photosynthesis antenna proteins. Lastly, for EBR VS Brz, the enrichment terms included glutathione metabolism, oxidative phosphorylation, and photosynthesis antenna proteins (Figure 3B).

Consistently with the above results, GO term enrichment analysis of DEPs for CK VS Brz, CK VS EBR, and EBR VS Brz indicated single-organism cellular processes, single-organism metabolic processes, photosynthesis, photosynthetic membranes, thylakoids, thylakoid membranes, chloroplast thylakoids, plastids, electron carrier activity, and structural constituents of the cytoskeleton. Specific terms of enrichment for CK VS Brz were defense response, iron–sulfur cluster assembly, lignin biosynthetic process, plastid stroma, nucleoside binding, and sucrose synthase activity. Meanwhile, for Brz VS EBR, the enrichment terms included starch metabolic process, starch biosynthetic process, plastid envelope, oxidoreductase activity, chlorophyll II binding, and rRNA binding. Finally, for CK VS EBR, the enrichment terms included photosystem II assembly, NADP metabolic process, pentose phosphate shunt, organelle sub-compartment, GTPase activity, ATPase activity, and oxidoreductase activity (Figure 3C).

### 2.4. Phosphoproteomic Changes in Mo17 in Response to BR and Brz Treatments

A total of 2202 phosphopeptides were identified in the MS data, while only 1699 phosphopeptide phosphoRS site probabilities were greater than 75%, and the phosphoRE binomial peptide score was greater than 50. Subsequently, the 1699 phosphopeptides were analysed. A threshold of FC (fold change) ≥ 1.2 for upregulated phosphopeptides and FC ≤ 0.833 for downregulated phosphopeptides was applied, together with a *p*-value ≤ 0.05. Based on these standards, 89 DEPs were identified in CK VS Brz, including 43 up- and 46 downregulated proteins. Similarly, for CK VS EBR, 92 DEPs were identified, of which 43 were up- and 49 were downregulated proteins. Lastly, 88 DEPs were identified between Brz and EBR, including 41 up- and 47 downregulated proteins (Figure 4A).

The enriched KEGG pathways for CK VS Brz were pyruvate metabolism and mRNA surveillance. In turn, for Brz VS EBR, KEGG pathway enrichment mainly indicated fructose and mannose metabolism, photosynthesis antenna proteins, spliceosome, and RNA transport. Finally, the enrichment terms for CK VS EBR were spliceosome and RNA transport (Figure 4B).

Additionally, the GO term enrichment analysis of CK VS Brz, CK VS EBR, and Brz VS EBR identified plastid inner members, nucleoside binding, and purine nucleoside binding. Specific terms of enrichment for CK VS Brz were electron carrier activity, photosynthesis, plastid thylakoid, plastid stroma, sucrose synthase activity, and NADH dehydrogenase activity. As for CK VS EBR, the GO enrichment terms were NADPH activity, sucrose transmembrane transporter activity, plastid ribosomes, and intrinsic components of the plastid membrane. Lastly, the enrichment terms for EBR VS Brz included the acetyl-CoA biosynthetic process, thylakoid membrane, photosystem I, starch binding, NADPH activity, and acetate-CoA ligase activity (Figure 4C).

### 2.5. Combined Transcriptome and Proteome Analyses of the Differences in the EBR and Brz Signaling Pathway in Maize Leaves

Combining DEGs and DEPs, we found that there were only 22 DEGs in the proteome and transcriptome data for CK VS Brz (Figure 5A). The GO enrichment analysis of DEGs included oxidoreductase activity, organic acid biosynthetic processes, protochlorophyllide reductase activity, UDP-glucose 6-dehydrogenase activity, and NAD(P)H dehydrogenase (quinone) activity. Seven genes involved in oxidoreductase activity were identified: *Zm00001d001820* (*pcr1*), *Zm00001d025103*, *Zm00001d034072* (*TIDP3299*), *Zm00001d042541 (LOX)*, *Zm00001d043249* (*csu690*), *Zm00001d048634 (bx6)*, and *Zm00001d048705*. Additionally, four genes involved in organic acid biosynthetic process were identified: *Zm00001d009354*, *Zm00001d013644* (*TIDP2999*), *Zm00001d028750* (*asn3*), and *Zm00001d042541* (*LOX*). As for CK VS EBR, 16 DEGs were identified in both proteome and transcriptome data. The GO enrichment analysis of DEGs and DEPs included oxidoreductase activity, the oxidation–reduction process, transaminase activity, the chlorophyll biosynthetic process, NADPH binding, the aspartate family amino acid metabolic process, protochlorophyllide reductase activity, and chromatin assembly. Additionally, four genes involved in oxidoreductase activity were identified: *Zm00001d001820* (*pcr1*), *Zm00001d015618*, *Zm00001d016185*, and *Zm00001d040173* (*irl1*) (Figure 5B). Meanwhile, *Zm00001d001820* (*pcr1*) was associated with the chlorophyll biosynthetic process and NADPH binding. As for the EBR VS Brz group, 18 DEGs were identified by combining proteome and transcriptome data. The KEGG enrichment analysis mainly indicated phenylalanine metabolism, porphyrin and chlorophyll metabolism, selenocompound metabolism, and photosynthesis antenna proteins (Figure 5C). One gene involved in photosystem I and photosystem II, *Zm00001d050403*, participates in photosynthetic light harvesting. Additionally, Zm00001d039270 regulates chloroplast localization and relocation. Finally, only one gene participates in phosphorylation: *Zm00001d021836*.

### 2.6. Combined Transcriptome and Phosphoproteome Analyses of the Differences in the EBR and Brz Signaling Pathway in Maize Leaves

Combining the DEGs and DEPs identified by transcriptome and phosphoproteome analyses, we found only three, seven, and four genes in the CK VS EBR, CK VS Brz, and EBR VS Brz groups, respectively (Figure 6A). The GO enrichment of DEGs included NAD binding, cytokinesis, nucleotide binding, sulfate transport, ATP binding, and cell division (Figure 6B). Three genes involved in ATP binding were identified, namely, *Zm00001d027934* (*ltk1*), *Zm00001d033777*, and *Zm00001d048924*. In turn, four genes were enriched for nucleotide binding: *Zm00001d013245* (*gpm540b*), *Zm00001d027934* (*ltk1*), *Zm00001d033777*, and *Zm00001d048924*. The MAP family protein Zm00001d020685 is involved in cytokinesis, microtubule cytoskeleton organization, cell division, and microtubule binding. As for the CK VS EBR group, we found only three DEGs, based on the transcriptome and phosphoproteomic data, i.e., CER5 (Zm00001d010426), transmembrane amino acid transporter family protein (Zm00001d004340), and ubiquitin-specific protease family C19-related protein (Zm00001d026258). CER5 is an ATP-binding cassette (ABC) superfamily protein associated with the ABC transporter complex. Meanwhile, only four genes were identified in the EBR VS Brz group, based on the transcriptome and phosphoproteomic data. The GO enrichment analysis mainly indicated photosystems and responses to abiotic stimuli. The photosystem II light-harvesting complex gene (Zm00001d033132) participates in the photosynthesis antenna protein pathway. Meanwhile, dhn2 (Zm00001d051420) is mainly involved in the response to cold and water deprivation (Figure 5B). The KEGG enrichment analysis mainly indicated photosynthesis antenna proteins, ascorbate and aldarate metabolism, pentose and glucuronate interconversions, and amino sugar and nucleotide sugar metabolism (Figure 6C).

### 2.7. BR Signal-Responsive Transcription Factors (TFs)

Transcription factors (TFs), including photosystems, reportedly play important roles in plant growth and development [49]. The identification of TF genes can provide a solid foundation for the study of the molecular mechanisms underlying BR signaling in photosystems. We identified 42 TF-related genes (21 down- and 21 upregulated) and 186 TF-related genes (123 down- and 63 upregulated), including 20 gene families that were differentially expressed in CK VS EBR and CK VS Brz, respectively (Figure 7).

The WRKY family of TFs is one of the largest families of transcriptional regulators in higher plants and is involved in plant growth, development, and responses to abiotic stress. The *Zm00001d043025* and *Zm00001d044171* genes, belonging to the WRKY family, were regulated in both the CK VS EBR and the CK VS Brz groups. *Zm00001d043025* and *Zm00001d044171* are involved in the MAPK signaling pathway. WRKY TFs are involved in the mitogen-activated protein kinase (MAPK) signaling pathway, which is involved in stress-induced defensive responses. Meanwhile, *Zm00001d030028* (*myc7*) and *Zm00001d053162* (*bZIP110*), which were downregulated in the CK VS Brz group, are also involved in the MAPK signaling pathway.

### 2.8. Analysis of Photosynthesis Antenna Protein Responses to BR Signaling

Transcriptome analysis of the photosynthesis antenna proteins showed that two genes were downregulated in the CK VS Brz group: *Zm00001d009589* (*lhcb3*) and *Zm00001d011285* (*lhcb10*). Meanwhile, only one gene was downregulated in CK VS EBR: *Zm00001d001857* (light-harvesting complex photosystem II subunit 6), and three genes were upregulated, while one gene was downregulated in the EBR VS Brz group, i.e., *Zm00001d001857* (light-harvesting complex photosystem II subunit 6), *Zm00001d050403* (chlorophyll a-b binding protein 4), *Zm00001d033132* (lhcb12, photosystem II light-harvesting complex gene 2.1), and *Zm00001d011285* (lhcb10) (Figure 8B). 

The proteome analysis of the photosynthesis antenna proteins identified only Zm00001d050403 (chlorophyll a-b binding protein 4) in the CK VS Brz group. Additionally, three proteins were identified in the EBR group: Zm00001d050403 (chlorophyll a-b binding protein 4), Zm00001d007267 (lhcb5), and Zm00001d011285 (lhcb10). Meanwhile, we identified four proteins in the CK VS EBR group: Zm00001d021906 (lhca2, chlorophyll a-b binding protein), Zm00001d046786 (Photosystem I chlorophyll a/b-binding protein 3-1 chloroplastic), Zm00001d050403 (chlorophyll a-b binding protein 4), and Zm00001d007267 (lhcb5) (Figure 8C).

## 3. Discussion

### 3.1. The Transcriptome Analysis Did Not Fully Match the Proteome and Phosphoproteome Analysis

High-throughput sequencing technology, as a fast and efficient platform, has been widely used to characterize internal changes and discover new genes in many species [50,51,52]. However, the transcriptome cannot fully represent the proteome, and transcription represents only half of the story [53]. Multiple omics analyses account better for the real situation [42]. In this study, we detected 39,591 transcription products, 3466 proteins, and 1066 phosphoproteins. However, more than 70% of the DEPs were not detected by transcriptome analysis, and only 2.9% of the differentially expressed phosphoproteins were differentially expressed as per transcriptome analysis. These results showed that post-translational regulation and protein modification may also be important in protein production.

### 3.2. Role of TFs in Maize Responses to BR Signaling

Plant responses to BR signaling involve a complex regulatory network of TFs and other regulatory genes that control enzymes, regulatory proteins, and metabolites. In previous studies, AtWRKY46, AtWRKY54, and AtWRKY70 were found involved in BR regulation of plant growth and development [54]. WRKY TFs are involved in the MAPK signaling pathway, which is involved in BR responses [55]. In this study, 30 WRKY genes were regulated in the CK VS Brz group, of which *Zm00001d043025* and *Zm00001d044171* are involved in the MAPK signaling pathway. AtWRKY33, an ortholog gene of *Zm00001d043025* in Arabidopsis, is a key transcriptional regulator that interacts with MPK3/6 to participate in the BR and other hormonal signaling pathways [56,57,58]. TR OsWRKY12, an ortholog gene of *Zm00001d044171* in rice, is involved in stress responses during the early vegetative stage, whereas BR plays an important role in stress responses [59]. In turn, *Zm00001d043025* and *Zm00001d044171* may modulate BR signaling in leaves by regulating target genes and interacting with MPK in the MAPK signaling pathway.

### 3.3. Response of Photosynthesis Antenna Proteins to BR Signaling

Brassinosteroids are some of the most important hormones in photosynthesis. The leaves are extraordinarily sensitive to environmental changes, and their photosynthetic performance is easily influenced [60]. Several studies demonstrated that exogenously applied BR can regulate photosynthesis, using diverse methods, such as foliar spray [31], seed soak [61], leaf segment soak [62], and shoot soak [63]. Further, BR effects on photosynthesis have multiple aspects and occur at multiple levels, as they may influence CO_2_ fixation rate [64], ribulose-1,5-bisphosphate (RuBP) regeneration and activity [65,66], chlorophyll a/b and carotenoids content [67], and chloroplast ultrastructure [68]. Photosynthetic antenna proteins are important for photosynthesis [69]. In previous studies, Zm00001d009589 (lhcb3) was shown to be involved in chloroplast development. Additionally, Zm00001d009589 (lhcb3) and Zm00001d011285 (lhcb10) respond to drought resistance [70,71]. BR can regulate plant growth and drought responses through the WRKY TF family [54]. In this study, Zm00001d009589 (lhcb3) was confirmed to be regulated by BR, and 30 WRKY TF family members were found to regulate BR. We speculate that BR also regulates the expression of Zm00001d009589 (lhcb3) and Zm00001d011285 (lhcb10) via the WRKY TF family in maize leaves. 

Chlorophyll a-b binding proteins play an essential role in absorbing light and transferring energy [72]. In previous studies, Zm00001d033132 (lhcb12) was reported to be involved in the photosynthetic system response to low nitrogen supply and to be important for chlorophyll fluorescence [73]. Zm00001d021906 (lhca2) is important for maize to improve photosynthesis under abiotic stresses, such as drought and low temperature [74,75]. Here, we found that the Zm00001d033132 (lhcb12) and Zm00001d021906 (lhca2) protein content were regulated by BR. In addition, the *Zm00001d050403* (chlorophyll a-b binding protein 4) transcription levels were downregulated, whereas its translation levels were upregulated in the CK VS EBR group. We hypothesize that BR regulates chlorophyll a-b binding protein both at the transcriptional and at the translational level and that the corresponding transcription and translation regulatory pathways are independent.

## 4. Materials and Methods

### 4.1. Plant Materials and Hormone Induction Procedure

The inbred maize line Mo17, provided by the College of Agronomy, Sichuan Agricultural University, China, was grown in growth cabinets. The seeds were germinated on a moist filter paper at 28 °C for two days. Then, the sprouted seeds were grown into seedlings, raised at 25 °C with a 16/8 h day/night photoperiod. Two fully developed leaves per seeding were used for hormone treatment. All seedlings were randomly divided into three groups, each with seven seedlings. One group was induced with 0.5 μM EBR (epibrassinolide), and the other was induced with 4 μM Brz (brassinazole). The leaves were induced with ddH_2_O as a mock control. The mock-, BR-, and Brz-treated leaves were harvested after 24 h of induction, immediately frozen in liquid nitrogen, and stored at −80 °C until use. Every group of treated leaves was collected and divided into five tubes: three for transcriptome analysis, one for proteome analysis, and one for phosphoproteomic analysis. 

### 4.2. Transcriptome Library Construction, Sequencing, and Annotation

Total RNA was extracted from each sample using the TRIzol reagent (Invitrogen, Carlsbad, CA, USA). The concentration and quality of the RNA samples were determined using an Agilent 2100 Bioanalyzer (Santa Clara, CA, USA). Poly T oligo-attached magnetic beads were used to purify poly A-containing mRNA molecules. The purified mRNA was broken into approximately 300 nt and was reverse-transcribed into first-strand cDNA using 6 nt random primers, and then the second-strand cDNA was synthesized using first-strand cDNA as a template. After library construction, PCR was used to enrich the fragments at approximately 450 nt in the library. The library was qualified using an Agilent 2100 Bioanalyzer, and the library concentration and effective concentration were determined. The libraries were sequenced on a HiSeq sequencer (Illumina) using the paired-end (PE) method.

The sequences were aligned to the *Zea mays* B73 reference genome B73 V4, using HISAT2 (hierarchical indexing for spliced alignment of transcripts) [76]. Cutadapt and FastQC were used to control the data quality. DEGs were identified using DESeq [77]. Meanwhile, the DEGs were defined based on *p*-values < 0.05, and the up-or downregulated genes were identified among the DEGs based on a log2 fold change > 1 or <−1, respectively.

Gene ontology (GO) and Kyoto Encyclopedia of Genes and Genomes (KEGG) pathways were assigned to the DEGs using Blast2GO. GO enrichment analysis was performed using topGO software. A GO term was considered significantly enriched if the *p*-value was <0.05. The DEGs were mainly involved in biological functions. 

### 4.3. Proteome Sequencing and Data Analysis

All samples were homogenized in liquid nitrogen. The powder was transferred to a pre-cooled 1.5 mL centrifuge tube with 1 mL of 10% TCA/cold acetone (with 65 mM DTT). The tube was then kept at −20 °C for 2 h. The extract was then separated by centrifuging (12,000× *g*) for 45 min, and the supernatant was discarded. This step was repeated three times. The proteins were vacuum-freeze-dried and resuspended in SDT buffer (4% SDS, 100 mM Tris-HCL, 100 mM DTT, pH 8.0). The supernatants were then incubated at 100 °C for 10 min, and the proteins were lysed by sonication on ice for 5 min (25 W, sonication 3 s, interval 7S). After boiling, the proteins were centrifuged at 14,000× *g* for 30 min, and the supernatants were collected. Protein concentration was evaluated using a BCA kit (Beyotime, Shanghai, China). 

The supernatant from each sample, containing precisely 300 µg of protein, was reduced with 100 mM DTT. For protein digestion, trypsin (Promega, Madison, WI, USA) was added at a 1:50 trypsin-to-protein mass ratio for 18 h of digestion. The digested solutions were desalted and dried via vacuum centrifugation. The desalted peptides were labeled with the TMT6plex^TM^ Isobaric Mass Tagging Kit (Thermo Scientific) following the manufacturer’s instructions. 

The peptides were dissolved in solvent A (0.1% formic acid) and separated in a gradient of solvent B (0.1% formic acid in 98% acetonitrile), 5–28% solvent B for 40 min, 28–90% solvent B for 2 min, and 90% solvent B for 18 min. After desalting and separating, the tryptic peptides were analyzed using tandem mass spectrometry (MS/MS, Orbitrap-Elite, Thermo Finnigan, San Jose, CA, USA). A full mass scan over the range of 350–2000 *m/z* was obtained with a resolution of 60,000, while the fragments were detected in the Orbitrap at a resolution of 15,000 at *m/z* 100. The normalized collision energy applied was 35 eV, and the automatic gain control (AGC) was set to 5E4 according to Han et al. (2020) [77]. 

The proteome discoverer 2.3 was applied to process the MS/MS data. The tandem mass spectra of the peptides were searched using the Zea_mays_AGPv4_pep.fasta database. The false discovery rata (FDR) was adjusted to <0.01, and the maximum number of missed cleavages was two. Peptide mass tolerance was set at ±20 ppm.

The MS raw data for phosphopeptide enrichment were analyzed using Mascot 2.3 and Proteome Discover 2.3 software (Thermo Scientific). The data were searched using the Zea_mays_AGPv4_pep.fasta database and the Mascot 2.3 software. Phosphopeptides were identified with Proteome Discover 2.3 software, which includes the Phospho RS and Phospho RS site probability analysis. Usually, the reliability of phosphorylation modification is higher if the Phospho RS score is >20 and the Phospho RS site probabilities are >75% [78,79].

### 4.4. Phosphoprotein Sequencing and Data Analysis

For phosphopeptide enrichment, the peptides in each fraction were dried by vacuum centrifugation. Then, 1×DHB buffer was used to dissolve the peptides. The phosphopeptides absorbed by the TiO_2_ beads were collected by centrifugation and washed with washing buffer1 and washing buffer2 in the kit. The enriched phosphopeptides were eluted with elution buffer and dried by vacuum centrifugation for liquid chromatography–tandem mass spectrometry (LC-MS/MS) analysis. Data analysis was consistent with that of the proteome.

## 5. Conclusions

BR signaling during photosynthesis resulted in genes, proteins, and phosphorylated proteins related to photosynthesis antenna proteins being significantly differentially expressed in both the CK VS EBR and the CK VS Brz groups. This study confirmed known BR signal-related genes and identified new putative BR-related genes. Additionally, the study revealed that translational and post-transcriptional translation under BR signaling plays an important role in photosynthesis. Overall, our results identified new TFs and photosynthesis antenna proteins involved in BR-mediated regulation of the photosynthesis pathway.

## Figures and Tables

**Figure 1 plants-12-01290-f001:**
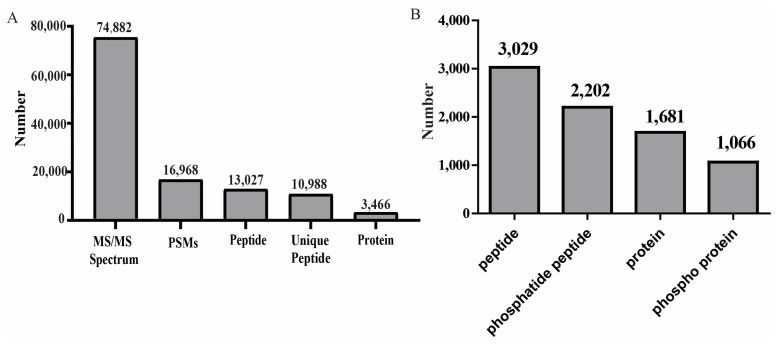
The numbers of identified peptides were statistically analyzed. (**A**) Number of the different peptides identified in the proteome data. (**B**) Number of the different peptides identified in the phosphoproteomic data.

**Figure 2 plants-12-01290-f002:**
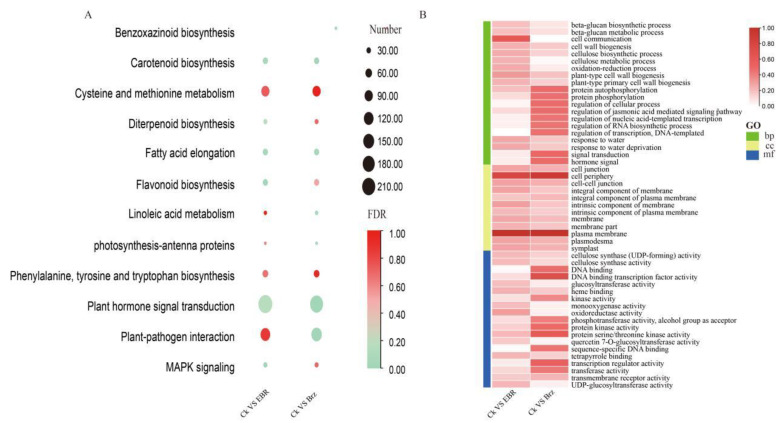
Enrichment analysis of BR-responsive transcriptome changes in maize leaves. (**A**) KEGG enrichment analysis of DEGs shown in a bubble chart. (**B**) GO enrichment analysis of DEGs shown in a heat map.

**Figure 3 plants-12-01290-f003:**
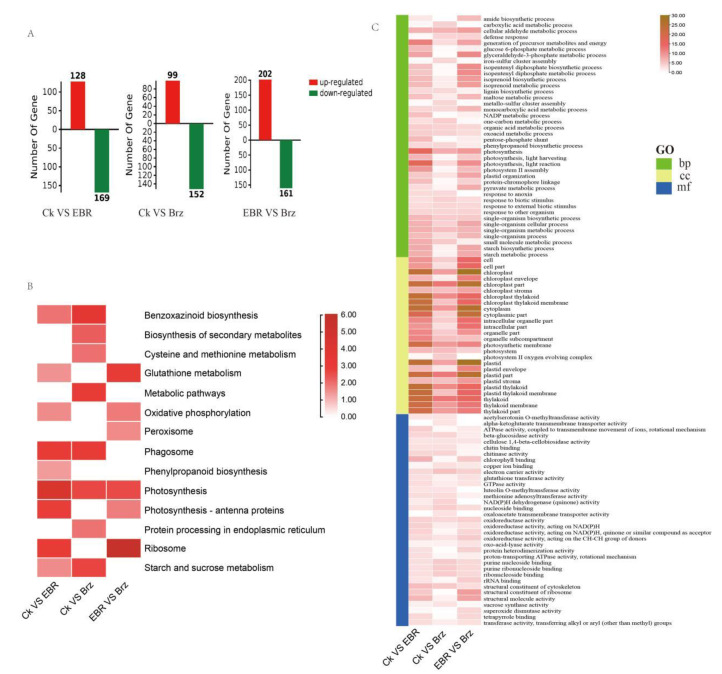
Enrichment analysis of BR-responsive proteome changes in maize leaves. (**A**) Statistical analysis of the number of DEPs. (**B**) The KEGG enrichment analysis of DEPs is shown as a heat map. (**C**) The GO enrichment analysis of DEPs is shown as a heat map.

**Figure 4 plants-12-01290-f004:**
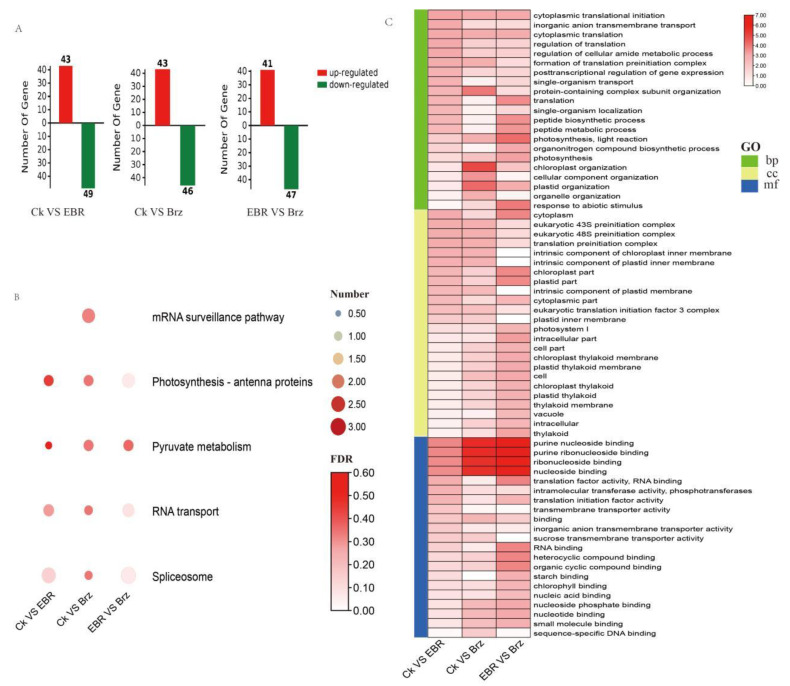
Enrichment analysis of the BR-responsive phosphoproteomic changes in maize leave. (**A**) The number of differentially expressed proteins was analyzed statistically. (**B**) KEGG enrichment analysis of differentially expressed proteins presented as a heat map. (**C**) GO enrichment analysis of differentially expressed proteins as a heat map.

**Figure 5 plants-12-01290-f005:**
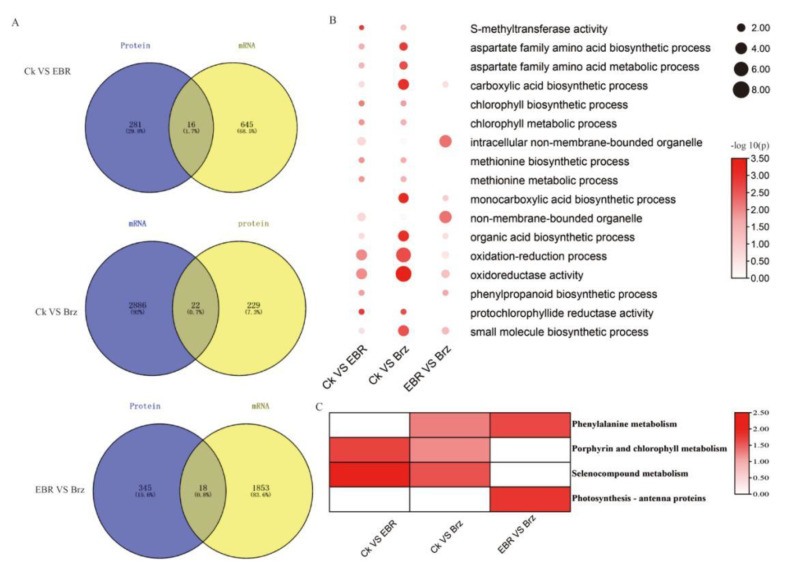
Enrichment analysis of the BR-responsive combined transcriptome and proteome changes in maize leaves. (**A**) The Venn diagrams of differentially expressed proteins were analyzed statistically. (**B**) The GO enrichment analysis of DEGs is shown as a heat map. (**C**) The KEGG enrichment analysis of DEGs is shown as a heat map.

**Figure 6 plants-12-01290-f006:**
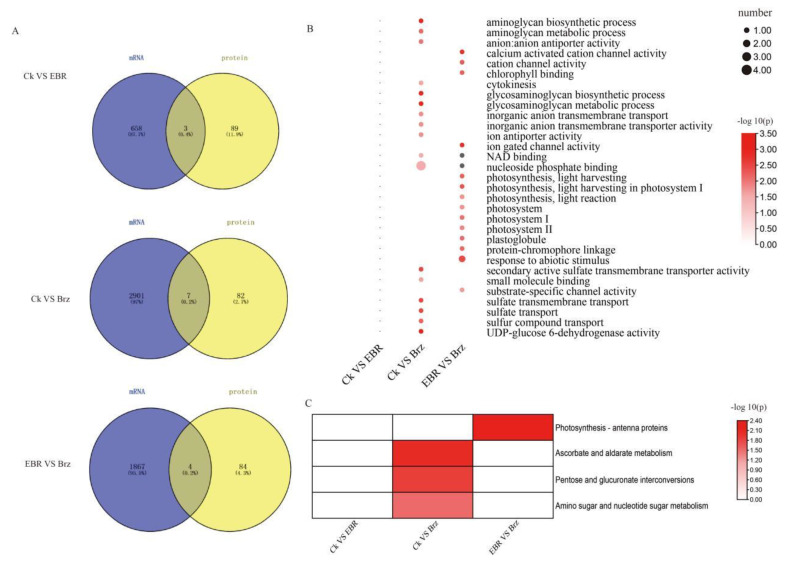
Enrichment analysis of the BR-responsive combined transcriptome and phosphoproteome changes in maize leaves. (**A**) The Venn diagrams of DEPs were analyzed statistically. (**B**) The GO enrichment analysis of the DEPs is shown as a heat map. (**C**) The KEGG enrichment analysis of DEPs is shown as a heat map.

**Figure 7 plants-12-01290-f007:**
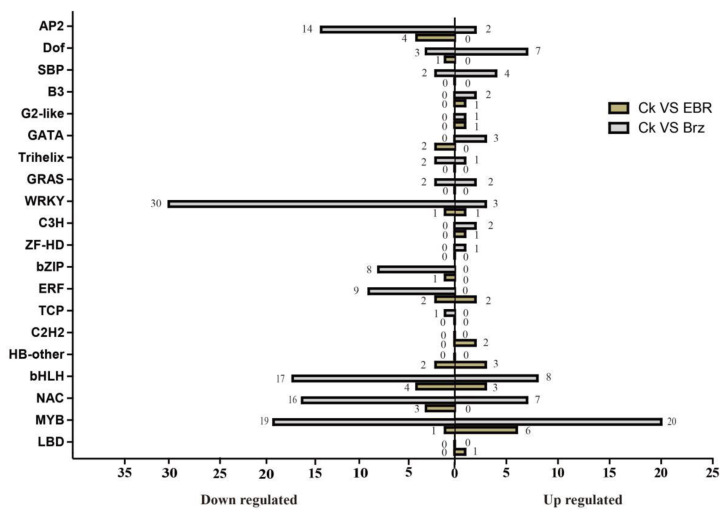
Identification of BR signal-responsive TFs.

**Figure 8 plants-12-01290-f008:**
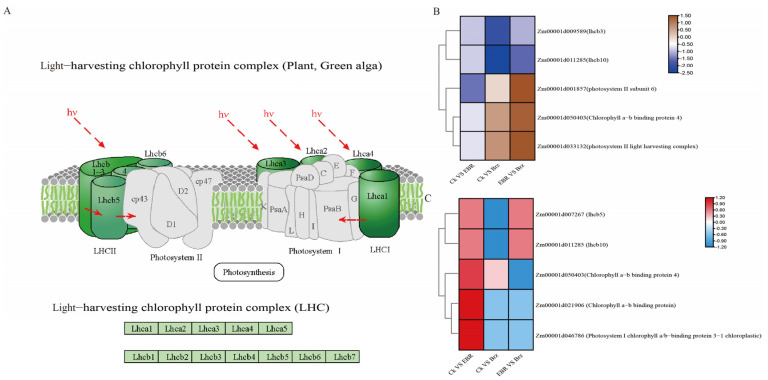
Differences in photosynthesis antenna proteins expression as per the transcriptome and proteome analyses. (**A**) Light-harvesting chlorophyll protein complex in the KEGG pathway database. (**B**) DEGs for photosynthesis antenna proteins as per transcriptome analysis. (**C**) DEPs regarding photosynthesis antenna proteins as per proteome analysis.

**Table 1 plants-12-01290-t001:** Mapped sequence reads from RNA-seq analysis.

Sample	No. of Reads	HISAT2	
		No. of Mapped Reads	% of Mapped Reads
CK1	47,265,794	40,808,687	86.34
CK2	36,858,382	32,171,708	87.28
CK3	37,525,170	32,802,893	87.42
EBR1	37,779,118	32,877,026	87.02
EBR2	36,097,508	31,370,499	86.90
EBR3	42,883,904	37,520,436	87.49
Brz1	43,591,944	38,366,049	88.01
Brz2	45,891,392	40,287,334	87.79
Brz3	39,977,534	35,050,577	87.68

**Table 2 plants-12-01290-t002:** Number of DEGs detected in comparison with other experiments.

Control	Treat	Up-Regulated	Down-Regulated	Total
Ck	EBR	128	169	297
Ck	Brz	99	152	251
EBR	Brz	202	161	363

## Data Availability

Not applicable.

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
