# Peer review of "Integrative Analysis of Transcriptome, Proteome, and Phosphoproteome Reveals Potential Roles of Photosynthesis Antenna Proteins in Response to Brassinosteroids Signaling in Maize"

_plants, 2023, doi:10.3390/plants12061290_

Round 1
Reviewer 1 Report
The research presented in the manuscript entitled “Integrative Analysis of Transcriptome, Proteome, and Phosphoproteome Reveals Potential Roles of Photosynthesis Antenna Proteins in Response to Brassinosteroids Signaling in Maize” by Hui Li, Xuewu He, Yuanfen Gao, Wenjuan Liu, Jun Song, and Junjie Zhang concerns the process of photosynthesis which is one of the most important processes that take place in the organelles of plant cells i.e. in the chloroplasts. In turn, brassinosteroids are phytohormones, which are considered analogous to animal steroid hormones in structure. They are involved in the regulation of different physiological processes connected with growth e.g. root growth, cell division and cell elongation. Moreover, they participate in morphogenesis, differentiation, seed germination and plant reproduction processes. As emphasized by the authors of the reviewed paper, this group of substances promote plant growth and productivity. The range of physiological processes that are regulated with brassinosteriods is wide, and I will not detail them here. In the reviewed paper, the authors of the research focused on the integrated transcriptome, proteome, and phosphoproteomic analyses to identify the key photosynthesis pathway that responds to brassinosteroid signaling. The object of study, plant species Zea mays, is known as one of the most important crop plant in the world.
In the first part of the paper, the “Introduction,” the authors inform the readers about some aspect concerning the process of photosynthesis. Then, they emphasized that brassinosteroids regulate photosynthesis both transcriptionally and by post-transcriptional modifications. The last part of “Introduction” gives some data concerning the technologies they used to performed the research. At the end, the authors give some methodological data which directly concerns the analysis performed during the research presented in the reviewed paper.
The results (part “2. Results”) of the research contain the effects of the analyses carried out using appropriate methods. They are summarized mainly in the detailed and extensive eight figures and in one table. All obtained results are described in detail. This part is the most extensive of all the parts included in the publication.
In the short part “3. Discussion” the results obtained by the authors are thoroughly analyzed and discussed using the data obtained in the results section. The results obtained by the authors are compared to the literature data in a manner typical for scientific papers on the presented subject or discipline.
In the next, fourth part of the publication, entitled “4. Materials and Methods” the authors describe the material (inbred maize line Mo17), how they treated it, and the methods which were used during study. The methods are collected in 3 subsections. The authors conducted the research in the manuscript using an appropriate number of techniques. These methods are properly selected by them. Moreover, the methods used by the authors are modern.
The results obtained by the authors are summarized in the conclusions section. This, fifth part of the paper is short, concise and adequate to the results obtained during the study. I would like also to emphasize the big number of citations used by the authors – 73.
To sum up, I am convinced that the reviewed paper is suitable for publishing in “Plants”.
Author Response
Thanks for your comments and recognition.
Reviewer 2 Report
The manuscript provides a comprehensive study of the molecular mechanism underlying the photosynthetic response to brassinosteroid signaling in maize. The authors employed transcriptome, proteome, and phosphoproteomic analyses to identify key pathways and genes that respond to brassinosteroid signaling. The study showed that photosynthesis-antenna proteins and carotenoid biosynthesis, plant hormone signal transduction, and MAPK signaling were significantly enriched in the list of differentially expressed genes upon brassinosteroid treatment. Furthermore, the study identified major genes and proteins that participate in photosynthesis antenna proteins that were upregulated by brassinosteroid treatment in a dose-dependent manner.
The study's strength lies in its comprehensive approach, which incorporates transcriptome, proteome, and phosphoproteomic analyses to identify key pathways and genes. Additionally, the study provides valuable information for a better understanding of the molecular mechanism underlying the photosynthetic response to brassinosteroid signaling in maize.
However, the manuscript could benefit from some improvements. First, the introduction could be expanded to provide more context for the study, particularly for readers who may not be familiar with brassinosteroids and their role in plant growth and productivity, the authors could provide more details on the methods used in the study, particularly on the transcriptome, proteome, and phosphoproteome analyses. Finally, the discussion could be expanded a little bit to provide more insights into the significance of the study's findings and their implications for future research.
Overall, the manuscript provides a valuable contribution to the understanding of the molecular mechanism underlying the photosynthetic response to brassinosteroid signaling in maize.
Below are some suggestions before the manuscript goes further:
1. Line 19, CK or Ck, VS. or V.S. please keep consistency for the whole manuscript. In the abstract BRZ (a BR biosynthesis inhibitor), BRZ or Brz keep consistent for the whole manuscript. For epibrassinolide (EBR), the author describes in line 42, it is the most active member among brassinosteroids, BRs), it is confused, as Epibrassinolide is a plant hormone, also known as a brassinosteroid, that plays a role in regulating various physiological processes in plants. It is the hormones, the author described as a member that seems like a gene or protein, please modify accordingly.
2. In the introduction part, the author should include on paragraph about the general introduction of BR signaling as mentioned above. And further describe, how is it in maize. as there are 11 key TFs in maize BR signaling pathway, how the function of these members studied in maize should be described here.
3. All the tables should be in scientific style, three-line tables if preferred in scientific writing
4. The front in figure 1 is not the same as well as resolution
5. Line 234-243 as well as the whole manuscript, the gene name should be in italics, and why is the gene name lowercase?
6. Line 320-321 in the discussion part, the sentence is redundant and can be deleted.
7. All the figures need improved resolution, this might be because I get the reviewer version and the final version might be ok.
8. Check the ref. one by one
Author Response
Point 1: Line 19, CK or Ck, VS. or V.S. please keep consistency for the whole manuscript. In the abstract BRZ (a BR biosynthesis inhibitor), BRZ or Brz keep consistent for the whole manuscript. For epibrassinolide (EBR), the author describes in line 42, it is the most active member among brassinosteroids, BRs), it is confused, as Epibrassinolide is a plant hormone, also known as a brassinosteroid, that plays a role in regulating various physiological processes in plants. It is the hormones, the author described as a member that seems like a gene or protein, please modify accordingly.
Response 1: Thanks for your comments. We rechecked the manuscript and keep the CK, VS, and Brz consistent. Meanwhile, we have modify the sentence more clear.
Point 2: In the introduction part, the author should include on paragraph about the general introduction of BR signaling as mentioned above. And further describe, how is it in maize. as there are 11 key TFs in maize BR signaling pathway, how the function of these members studied in maize should be described here.
Response 2: Thanks for your comments. We have add the BR signaling pathway in paragraph 2.
Point 3: All the tables should be in scientific style, three-line tables if preferred in scientific writing.
Response 3: Thanks for your comments. We have modify the table to three-line style.
Point 4: The front in figure 1 is not the same as well as resolution.
Response 4: Thanks for your comments. We edited the picture again and did our best to ensure the clarity of the picture.
Point 5: Line 234-243 as well as the whole manuscript, the gene name should be in italics, and why is the gene name lowercase?
Response 5: Thanks for your comments. We check the manuscript again and modify the gene name.
Point 6: Line 320-321 in the discussion part, the sentence is redundant and can be deleted.
Response 6: Thanks for your comments. We have delete the redundant sentence.
Point 7: All the figures need improved resolution, this might be because I get the reviewer version and the final version might be ok.
Response 7: Thanks for your comments. We edited the picture by Adobe Illustrator software again and did our best to ensure the clarity of the picture.
Point 8: Check the ref. one by one.
Response 8: Thanks for your comments. We have checked the ref. one by one.